# AS-Quant: Detection and Visualization of Alternative Splicing Events with RNA-seq Data

**DOI:** 10.3390/ijms22094468

**Published:** 2021-04-25

**Authors:** Naima Ahmed Fahmi, Heba Nassereddeen, Jaewoong Chang, Meeyeon Park, Hsinsung Yeh, Jiao Sun, Deliang Fan, Jeongsik Yong, Wei Zhang

**Affiliations:** 1Department of Computer Science, University of Central Florida, Orlando, FL 32816, USA; fnaima@knights.ucf.edu (N.A.F.); jiao.sun@knights.ucf.edu (J.S.); 2Genomics and Bioinformatics Cluster, University of Central Florida, Orlando, FL 32816, USA; hebanasser@knights.ucf.edu; 3Department of Electrical and Computer Engineering, University of Central Florida, Orlando, FL 32816, USA; 4Department of Biochemistry, Molecular Biology and Biophysics, University of Minnesota Twin Cities, Minneapolis, MN 55455, USA; jwchang@umn.edu (J.C.); parkm@umn.edu (M.P.); yehxx099@umn.edu (H.Y.); 5School of Electrical, Computer and Energy Engineering, Arizona State University, Tempe, AZ 85287, USA; dfan@asu.edu

**Keywords:** alternative splicing, transcriptome, RNA-seq, RT-PCR, visualization

## Abstract

(1) Background: A simplistic understanding of the central dogma falls short in correlating the number of genes in the genome to the number of proteins in the proteome. Post-transcriptional alternative splicing contributes to the complexity of the proteome and is critical in understanding gene expression. mRNA-sequencing (RNA-seq) has been widely used to study the transcriptome and provides opportunity to detect alternative splicing events among different biological conditions. Despite the popularity of studying transcriptome variants with RNA-seq, few efficient and user-friendly bioinformatics tools have been developed for the genome-wide detection and visualization of alternative splicing events. (2) Results: We propose AS-Quant, (*A*lternative *S*plicing *Quant*itation), a robust program to identify alternative splicing events from RNA-seq data. We then extended AS-Quant to visualize the splicing events with short-read coverage plots along with complete gene annotation. The tool works in three major steps: (i) calculate the read coverage of the potential spliced exons and the corresponding gene; (ii) categorize the events into five different categories according to the annotation, and assess the significance of the events between two biological conditions; (iii) generate the short reads coverage plot for user specified splicing events. Our extensive experiments on simulated and real datasets demonstrate that AS-Quant outperforms the other three widely used baselines, SUPPA2, rMATS, and diffSplice for detecting alternative splicing events. Moreover, the significant alternative splicing events identified by AS-Quant between two biological contexts were validated by RT-PCR experiment. (3) Availability: AS-Quant is implemented in Python 3.0. Source code and a comprehensive user’s manual are freely available online.

## 1. Introduction

A single gene can contain multiple exons and introns in eukaryotes. Exons can be joined together by splicing in different ways. Recent studies have estimated that alternative splicing events exist in more than 95% of multi-exon genes in human and mouse [1,2,3], and it provides cells with the opportunity to create protein isoforms with multiple functions from a single gene. Alternative splicing is a central element in gene expression. It influences almost all aspects of protein functions, including binding between proteins and ligand, nucleic acids or membranes, localization and enzymatic properties [2]. Alternative splicing events are generally regulated by dynamic reorganization and binding of splicing factors to cis-sequences elements in pre-mRNA, and occurs in various fashions [1]. Figure 1 lists five major types of alternative splicing events [4] found in eukaryotes. They are: Skipped Exon (SE), Retained Intron (RI), Alternative 3’ Splice Site (A3SS), Alternative 5’ Splice Site (A5SS), and Mutually Exclusive Exon (MXE).

A precise detection of alternative splicing events among different biological contexts could provide insights into new molecular mechanisms and define high-resolution molecular signatures for phenotype predictions [5,6]. High-throughput RNA-seq platform is capable of studying splicing variants, and several bioinformatics tools have been developed to identify alternative splicing events with RNA-seq [4,7,8,9,10]. However, the selection for comprehensive and genome-wide assessments of the splicing events is limited, and few of the existing tools can provide high-resolution read coverage plots of the splicing events with accurate isoform annotation. We have developed AS-Quant, a program for genome-wide alternative splicing events detection and visualization. It efficiently handles large-scale alignment files with hundreds of millions of reads in different biological contexts and generates a comprehensive report for most, if not all, potential alternative splicing events (both annotated and unannotated), and also generates high quality plots for the splicing events.

## 2. Results

In the experiments, AS-Quant was compared with three widely used alternative splicing events detection pipelines, SUPPA2 [9], rMATS [8], and diffSplice [7] on both simulated and real RNA-seq datasets. In the simulation experiment, flux-simulator [11] was applied to generate synthetic RNA-seq data with true alternative splicing events between different samples. In the real RNA-seq experiment on mouse embryonic fibroblast (MEF) cell lines, the detected alternative splicing events were validated by RT-PCR experiment.

### 2.1. Experimental Results with Simulated RNA-seq Data

In the simulations, we applied flux-simulator [11] to generate paired-end short reads to simulate RNA-seq experiment in silico based on a ground truth transcript expression profile and list of alternative splicing events between two conditions, using mm10 reference mouse genome with UCSC annotation. 1565 alternative splicing events, separated in five categories (Figure 1) were generated between two conditions. For each condition, three replicates were created by repeating the whole experiment with the same parameter settings in the flux-simulator to represent the samples in two different biological conditions. The flux-simulator parameters used in this experiment are listed in the Appendix A. To generate the ground truth expression profile and the alternative splicing events, the gene expressions were sampled from a Poisson distribution to reflect real RNA-seq data [12]. For the genes with true splicing events, the transcripts in the gene were separated into two groups, one with spliced exon and one without spliced exon (for the genes with MXE event, the transcript(s) in each group contain one spliced exon). The expression proportions of the two groups in the same gene are set significantly different for the two conditions (i.e., the proportion difference was larger than 10%) to support the presence of altered exons. Whereas, for the gene without splicing events, the transcript expressions in the two conditions are kept similar to each other. In such way, we simulated 766, 108, 94, 368, and 229 alternative splicing events for the five categories, SE, MXE, RI, A3SS, and A5SS, respectively.

In the simulation experiment, 50 million paired-end reads with 76 bps of each end were generated for each replicate in each condition by flux-simulator. AS-Quant was compared with SUPPA2, rMATS, and diffSplice on the simulated RNA-seq data. The area under the ROC curve (AUC), sensitivity (true positive rate), and specificity (true negative rate) were calculated to evaluate the performance of AS-Quant and baseline methods. From the results reported in Table 1 and the ROC curves in Figure 2, we observed that AS-Quant has the best overall performance (AUC = 0.84) followed by SUPPA2 (AUC = 0.80) and diffSplice (AUC = 0.74). rMATS (AUC = 0.65) did not work very well compared to the other three methods due to a large number of false positive events. In addition, we also report the performance of the four methods on each type of alternative splicing events in Table 2 and the ROC curves in Appendix A. AS-Quant got the highest AUC scores in four out of five types. In three out of five categories (i.e., SE, MXE, A3SS), the AUC scores for AS-Quant were close to 1, which indicates that our pipeline can almost perfectly detect these three types of events in the simulated RNA-seq data. SUPPA2 requires the transcript quantification results from a RNA-seq quantification method (e.g., Salmon [13]) as its input, and its performance depends on the quality of the input transcript expressions. In this simulation experiment, SUPPA2 did better than AS-Quant on A5SS. rMATS got the highest sensitivity score on SE since it reported more positive events compared to the other three methods, but the specificity score was much lower than the other methods due to a large number of false ones in its reported events. Both AS-Quant and SUPPA2 got the highest specificity score in three out of five categories and diffSplice got the highest specificity score on MXE. Overall, AS-Quant outperformed SUPPA2, rMATS, and diffSplice on detecting true alternative splicing events in the simulation experiment.

To learn the impact of sequencing depths on analysis of alternative splicing with AS-Quant, we simulated several RNA-seq data with different read depths, i.e., 1 M (million), 2 M, 5 M, 10 M, 30 M, and 50 M paired-end reads by flux-simulator with the same parameter setting and ran AS-Quant on each experiment separately. In each experiment, three replicates were generated in each condition and the same 1565 alternative splicing events were generated as a ground truth between two conditions using the same procedures as mentioned in the previous section. The ROC curves for different read depth are shown in Figure 3. We observed that after the sequencing depth reached 5 million, the performance of AS-Quant are almost same. The result suggests that AS-Quant is relative robust for alternative splicing events detection on low read coverage samples and lowly expressed genes.

### 2.2. Experimental Results with Mouse Embryonic Fibroblasts (MEFs) Samples

To evaluate the performance of AS-Quant on the real RNA-seq data, two MEFs samples, $Tsc1^{-/-}$ MEFs with control and U2af1 knocked down (siU2af1) were used in the analysis (SRP215854). U2af1 is a splicing factor and plays a role in alternative splicing [14]. In total, 81,677,330 paired-end reads for $Tsc1^{-/-}$ control and 87,017,091 paired-end reads for siU2af1 were produced from Hi-Seq pipeline with length of 51 bp of each end. The raw RNA-seq fastq files were aligned to mouse mm10 reference genome using TopHat2 [15]. AS-Quant was then applied to detect significant alternative splicing events between the two samples with *p*-value < 0.05 and absolute ratio difference between the two samples > 0.1 (Equation (Equation 2)). AS-Quant detected 257, 5, 43, 101, and 30 events for SE, RI, MXE, A3SS, and A5SS, respectively. The three baseline methods, SUPPA2, rMATS, and diffSplice, were also applied on the two samples. The splicing events identified by SUPPA2 were specified based on the difference of relative abundances between two conditions, called percent spliced-in (PSI) value > 0.2 and *p*-value < 0.05. The significant events reported by rMATS were selected based on the two thresholds, |ΔΨ| > 5% (Ψ is the exon inclusion level) and false discovery rate (FDR) ≤ 1% [8], whereas the significant events identified by diffSplice were determined by the square root of the Jensen–Shannon divergence (JSD) > 0.2 [7]. Table 3 shows the number of events detected by the four methods in the five splicing categories.

Based on the potential events reported by AS-Quant, four genes, *Ptbp1*, *Ganab*, *Camk2g* and *Tpm3*, were selected for validation. These genes were selected due the design of PCR (polymerase chain reaction) primers for wet-lab validation. The alternative splicing events in *Ptbp1* and *Ganab* were also identified by SUPPA2, rMATS, and diffSplice, whereas the event in *Camk2g* was only identified by AS-Quant and the event in *Tpm3* was only identified by AS-Quant and SUPPA2. Figure 4a represents the two read coverage plots for *Ptbp1* and *Ganab*, and Figure 5a shows the read coverage plots for *Camk2g* and *Tpm3*. The alternative spliced exons are marked in yellow in the plots. RT (reverse transcription)-PCR and agarose gel electrophoresis were applied to validate the expression of the transcript isoforms with exon inclusion/exclusion. As shown in Figure 4b and Figure 5b, exon inclusion and exclusion between the two samples show significant changes, which is consistent with our observations on the RNA-seq read coverage plots reported in Figure 4a and Figure 5a. These results further confirmed that AS-Quant can identify not only the true alternative splicing events which can be detected by the baseline methods, but also the true events which are ignored by the baseline methods in the RNA-seq samples from two different biological contexts.

## 3. Discussion

The eukaryotic genome is capable of producing multiple isoforms from a gene by alternative splicing during pre-mRNA processing. It provides cells with the opportunity to create protein isoforms from the same gene to participate in different functional pathways. Therefore, accurately profiling transcript variants between different biological states could lead to the finding of new molecular mechanisms and potentially better molecular signals for phenotype predictions. However, the options for comprehensive and genome-scale assessment of the alternative splicing events are limited due to the low-resolution analyzing power. In this study we developed a novel and user-friendly pipeline, AS-Quant, to detect five different types of alternative splicing events between two biological contexts and provide high-resolution read coverage plots with annotation for user specified splicing events. The pipeline not only can accurately detect the splicing events compare to the baseline methods, but also provide reasonable running time. AS-Quant took 41 CPU minutes to run the experiments on the two MEFs samples, whereas SUPPA2, rMATS, and diffSplice took 29, 93, and 32 CPU minutes, respectively. The CPU time was measured on Intel(R) Xeon(R) CPU E5-2620 v4 @ 2.10 GHz. Although AS-Quant relies on current annotation, it can also detect unannotated SE event (Appendix A). Our pipeline will be further improved to do de
novo analysis for all five types of splicing events in future studies. Overall, this study reports an efficient and precise framework for isoform detection in the transcriptomic data.

## 4. Materials and Methods

AS-Quant workflow consists of three steps: (i) read coverage estimation of exons; (ii) alternative splicing events categorization and assessment; (iii) visualization of splicing events (Figure 6).

The first step requires aligned RNA-seq data of two different biological conditions in BAM format as the input. Each biological condition can contain multiple samples in each group. The read coverage files are generated from the annotated genes and exons for each sample with SAMtools [16]. In the second step, AS-Quant first identifies all potential alternative splicing events of five different categories (Figure 1) based on UCSC gene annotation following the lead of the study in [4] and the unannotated skipped exon (SE) events. The alternative splicing exon in each category is highlighted in yellow in the middle panel of Figure 6. For each spliced exon of each category, the average read coverage (*n*) is then computed by estimating the number of reads mapped to that exon ($r_{e}$) divided by its effective length ($l_{e}$).
(1)$$n=\frac{r_{e}}{l_{e}}$$

The average read coverage (*N*) for all other exons in that specific gene is calculated the same way, by dividing the number of mapped reads ($R_{\bar{e}}$) to all these exons by their summed length ($L_{\bar{e}}$). Next, the ratio differences between the two conditions are calculated based on the following equation:(2)$$\frac{n_{1}}{N_{1}}-\frac{n_{2}}{N_{2}},$$
where 1 and 2 represent the two conditions. A large positive ratio difference indicates a potential splicing event in condition 2, whereas a negative ratio difference with a large absolute value indicates a potential splicing event in condition 1. After that, a canonical 2 × 2 $\chi^{2}$-test is applied to report a *p*-value for each potential splicing event. If there is more than one sample or replicate under each condition, the Wilcoxon rank-sum test can be applied to determine the events. Only the events which show a significant *p*-value (<0.05) among the two conditions and a absolute ratio difference larger than a threshold (>0.1) will be reported.

Based on the significant splicing events reported in the second step, AS-Quant generates RNA-seq read coverage plots along with the gene’s annotation in the third step. For any user-specific input with chromosome and genomic coordinates (*chr:start_position-end_position*), AS-Quant illustrates the event with a high quality read coverage plot for a better illustration of the process. The bottom panel in Figure 6 shows an example of the read coverage plot generated from AS-Quant. Appendix A shows an example of the retained intron (RI) event detected by AS-Quant.

In this study, the area under the ROC curve (AUC), sensitivity (true positive rate) and specificity (true negative rate) were applied to evaluate the performance of AS-Quant and baseline methods. An ROC curve plots true positive rate and false positive rate at different classification thresholds and AUC provides an aggregate measure of performance across these classification thresholds. Sensitivity and specificity are defined as follows
(3)$$Sensitivity=\frac{TP}{TP\;+\;FN}$$
(4)$$Specificity=\frac{TN}{TN\;+\;FP}$$
where TP, TN, FP, and FN denote the number of true positives, true negatives, false positives, and false negatives, respectively. Three widely used alternative splicing events’ detection pipelines, SUPPA2 [9], rMATS [8], and diffSplice [7] were applied to compare the performance with AS-Quant. The command lines for running SUPPA2, rMATS, and diffSplice are available in the Appendix A.

## 5. Conclusions

We present AS-Quant, a computational pipeline that allows the identification of transcriptome-wide alternative splicing events in RNA-seq data. The significant events are illustrated by read coverage plots along with full annotations of a specific gene. The experimental results on two mouse MEFs samples by RT-PCR and simulated RNA-seq data demonstrate that AS-Quant is an accurate and efficient tool to detect alternative splicing events between samples with different biological contexts.

## Figures and Tables

**Figure 1 ijms-22-04468-f001:**
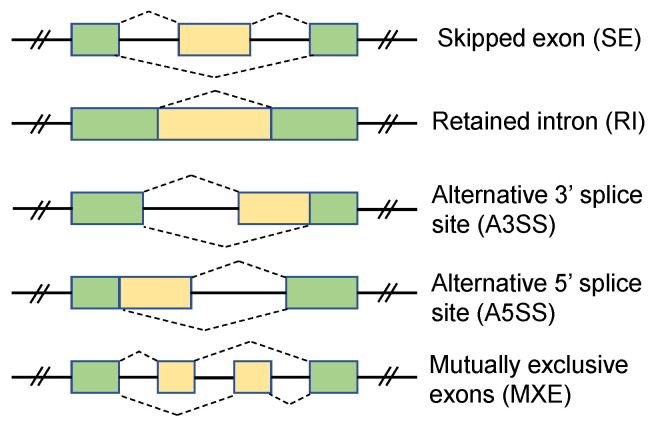
Five major types of alternative splicing events. The alternative splicing exon(s) in each category is highlighted in yellow.

**Figure 2 ijms-22-04468-f002:**
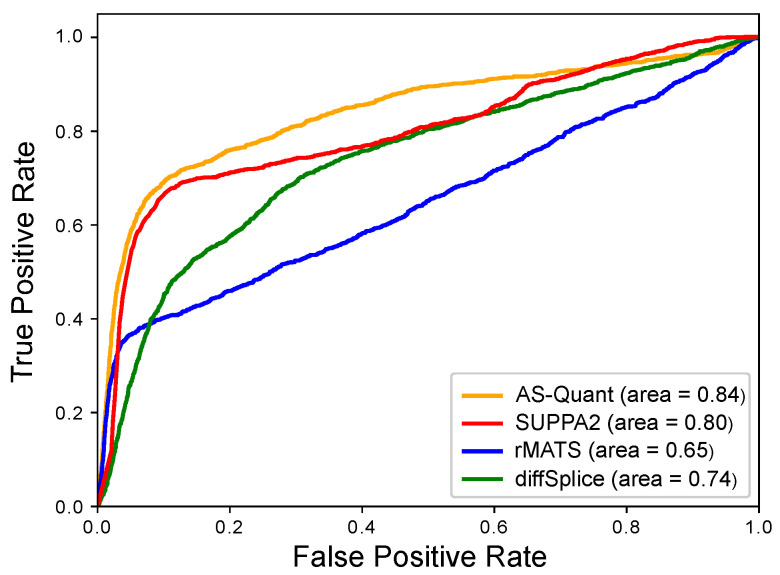
Simulation experiment to assess the performance of AS-Quant and baseline methods. The receiver operating characteristic (ROC) curves, i.e., true positive rate against false positive rate, are plotted.

**Figure 3 ijms-22-04468-f003:**
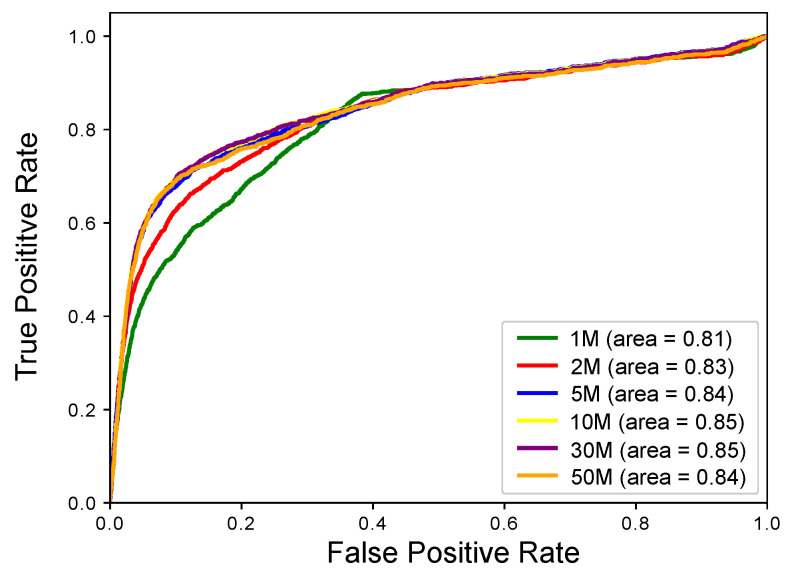
Simulation experiment to assess the performance of AS-Quant on different read depths. The ROC curves for the results of different RNA-seq read depth are plotted.

**Figure 4 ijms-22-04468-f004:**
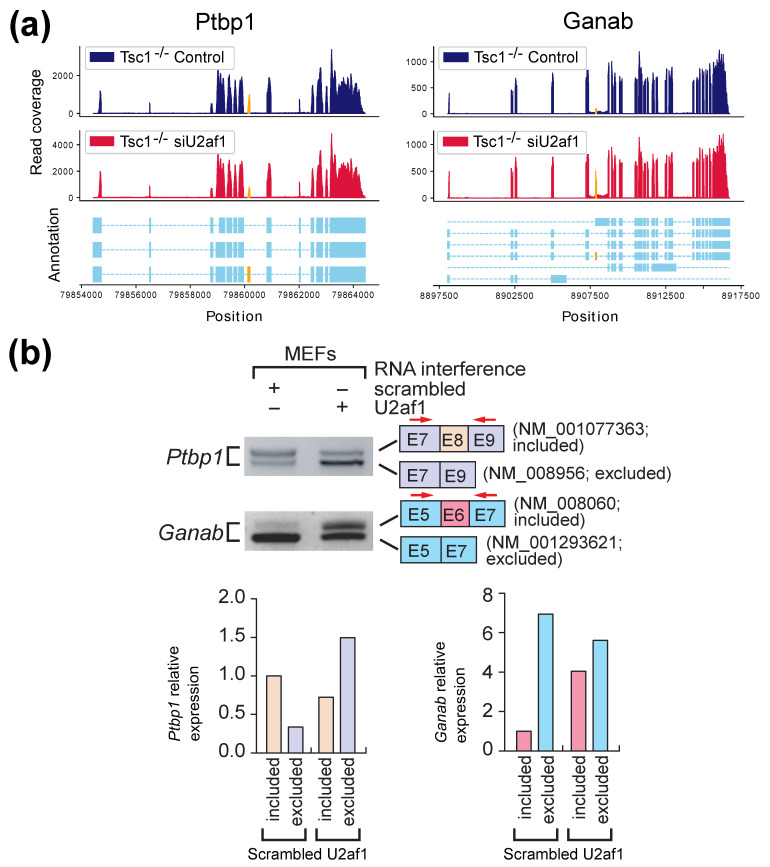
Validation of U2AF1-mediated alternative splicing events that are commonly detected by all four tested methods. (**a**) RNA-seq read coverage plots of the gene *Ptbp1* and *Ganab* in the two samples with accurate isoform annotations. Alternatively spliced exons are marked in yellow. (**b**) Validation of isoform expression using RT-PCR and agarose gel electrophoresis. Quantitation of gel images was done using ImageQuant software. Exon inclusion and exclusion are color-coded. Total RNAs from $Tsc1^{-/-}$ MEFs used for RNA-Seq experiments were used for RT-PCR amplification of *Ptbp1* or *Ganab* transcript isoforms. Scrambled RNA interference is control and U2af1 RNA interference is the case. The PCR primers to detect transcript isoforms for *Ptbp1* or *Ganab* were marked by red arrows and their sequences are reported in the Appendix A. Schematic of alternative spliced isoform structures for each PCR product is shown next to the gel image. Exon numbers and transcript identification numbers in RefSeq annotation are shown. A higher band intensity of PCR products indicates a higher expression of that specific transcript isoform.

**Figure 5 ijms-22-04468-f005:**
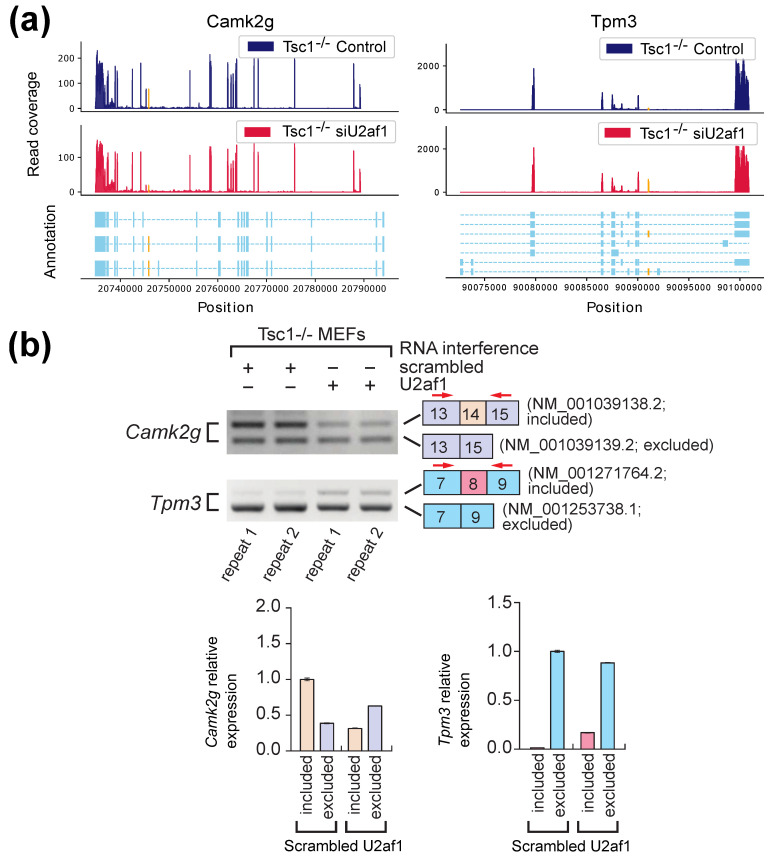
Validation of U2AF1-mediated alternative splicing events that are only detected by AS-Quant (*Camk2g*) or by AS-Quant and SUPPA2 (*Tpm3*). (**a**) Isoform structures of *Camk2g* and *Tpm3* gene and their RNA-seq read coverage plots. Alternatively spliced exons are marked in yellow. (**b**) Validation of isoform expressions was conducted using RT-PCR and agarose gel electrophoresis. Quantitation of gel images was done using ImageQuant software. Two biological repeats of experiment were performed. Exon inclusion and exclusion are color-coded. Total RNAs from $Tsc1^{-/-}$ MEFs used for RNA-Seq experiments were used for RT-PCR amplification of *Camk2g* or *Tpm3* transcript isoforms. Scrambled RNA interference is control and U2af1 RNA interference is the case. The PCR primers to detect transcript isoforms for *Camk2g* or *Tpm3* were marked by red arrows and their sequences are reported in the Appendix A. Schematic of alternative spliced isoform structures for each PCR product is shown next to the gel image. Exon numbers and transcript identification numbers in RefSeq annotation are shown. A higher band intensity of PCR products indicates a higher expression of that specific transcript isoform.

**Figure 6 ijms-22-04468-f006:**
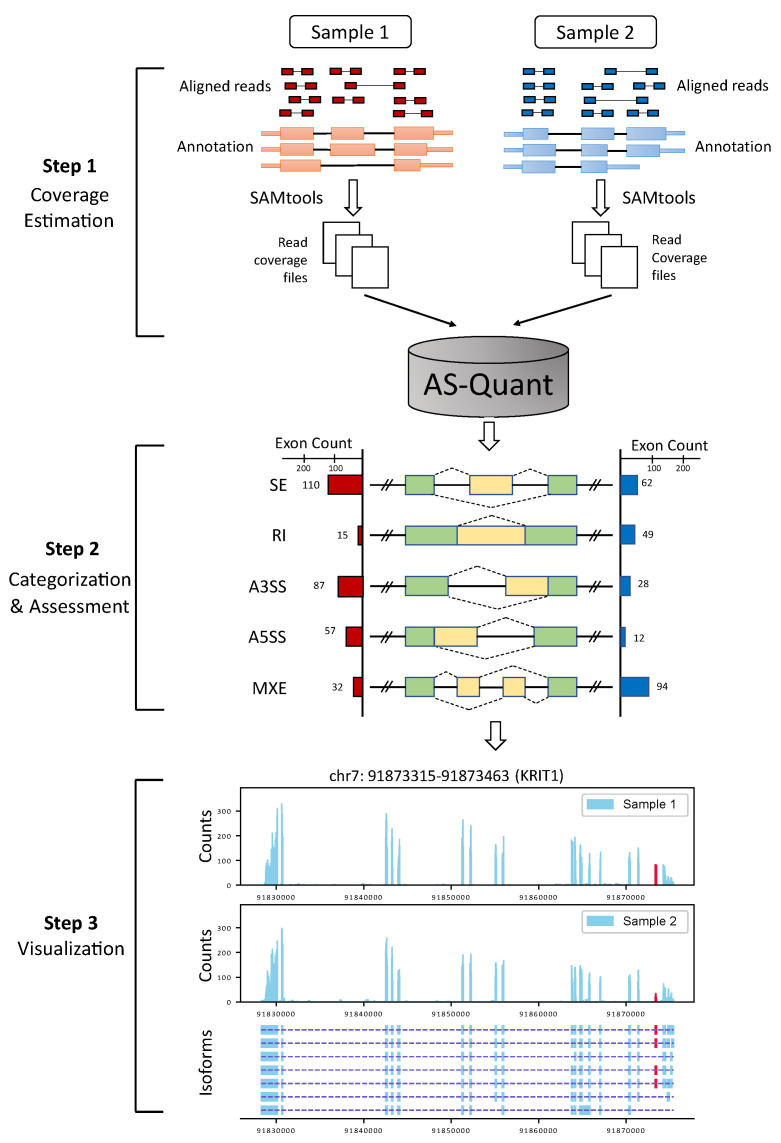
Workflow of AS-Quant. Starting with aligned RNA-seq bam files, AS-Quant consists of three steps (i) read coverage estimation, (ii) splicing events categorization and assessment, (iii) visualization.

**Table 1 ijms-22-04468-t001:** Comparison among AS-Quant, SUPPA2, rMATS and diffSplice on simulated RNA-seq data. AUC score, sensitivity, specificity of the four methods are reported. The best results across the four methods are bold.

Method	AUC	Sensitivity	Specificity
AS-Quant	**0.84**	**0.64**	**0.98**
SUPPA2	0.80	0.44	0.97
rMATS	0.65	0.22	0.49
diffSplice	0.74	0.05	0.79

**Table 2 ijms-22-04468-t002:** Comparison among AS-Quant, SUPPA2, rMATS, and diffSplice on simulated RNA-seq data. AUC score, sensitivity, specificity of the four methods on five different types of splicing events are reported. The best results across the four methods are bold.

AS Type	Method	AUC	Sensitivity	Specificity
	AS-Quant	**0.97**	0.60	**1.00**
SE	SUPPA2	0.84	0.64	0.99
	rMATS	0.84	**0.80**	0.50
	diffSplice	0.72	0.31	0.95
	AS-Quant	**0.78**	**0.31**	**1.00**
RI	SUPPA2	0.63	0.09	**1.00**
	rMATS	0.58	0.30	0.50
	diffSplice	0.53	0.01	0.98
	AS-Quant	**0.98**	**0.91**	0.82
MXE	SUPPA2	0.66	0.37	**1.00**
	rMATS	0.76	0.69	0.50
	diffSplice	0.46	0.03	**1.00**
	AS-Quant	**0.99**	**0.78**	**1.00**
A3SS	SUPPA2	0.80	0.56	0.99
	rMATS	0.49	0.58	0.50
	diffSplice	0.62	0.03	0.51
	AS-Quant	0.71	0.50	0.97
A5SS	SUPPA2	**0.83**	**0.66**	**0.99**
	rMATS	0.46	0.57	0.50
	diffSplice	0.58	0.03	0.51

**Table 3 ijms-22-04468-t003:** Number of alternative splicing events identified by AS-Quant and three baseline methods between $Tsc1^{-/-}$ MEFs with control and siU2af1. diffSplice cannot separate A3SS and A5SS.

	SE	RI	MXE	A3SS	A5SS
AS-Quant	257	5	43	101	30
SUPPA2	172	46	12	121	117
rMATS	1128	15	129	51	16
diffSplice	169	560	0	1125	

## Data Availability

Source code and a comprehensive user’s manual are freely available at https://github.com/CompbioLabUCF/AS-Quant (accessed on 23 April 2021). The mouse cell line data is available at SRP215854 (accessed on 23 April 2021).

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
