# Peer review of "AS-Quant: Detection and Visualization of Alternative Splicing Events with RNA-seq Data"

_ijms, 2021, doi:10.3390/ijms22094468_

Round 1

Reviewer 1 Report

In their manuscript "AS-Quant: Detection and Visualization of Alternative Splicing Events with RNA-seq Data" Fahmi et al. present a new Python-based tool "AS-Quant" for the detection and visualisation of alternative splicing (AS) events. They test the performance of their tool on simulated and real datasets and compare their results to those of established algorithms.

AS-Quant adds to a long list of existing AS detection tools but provides little novelty except for the built-in visualisation feature. 

Major points:
1. The manuscript is lacking references, which might have been lost during the submission process? This made the review of the manuscript difficult.

2. One major disadvantage is that AS-Quant can't detect non-annotated (novel) splicing events of which there are many. Most modern AS detection tools have this ability.

3. The performance of AS-Quant was compared to that of diffSplice and rMATS, two established tools that have been published in the years 2013 and 2014, respectively. There's been a lot of progress in the AS analysis field since these tools where published. To get an idea where AS-Quant stands in the landscape of current AS-detection methods the authors should compare AS-Quant's performance to 1-2 more recently published algorithms.
Examples are Whippet (PMID: 30220560), Psichomics (PMID: 30277515) or SUPPA2 (PMID: 29571299).

4. The statistical approach to identify significant differential alternative splicing events doesn't take advantage of the fact that most studies generate RNA-seq data in triplicates nowadays. I'd suggest to incorporate a more appropriate test for scenarios with n > 2 replicates.

5. The previous point could also improve AS visualisation in AS-Quant. Can the authors think of a way to show the variance in AS signals across replicates?

Minor points:

1. One way to make this tool more attractive would be if it would support long-read sequencing data.

2. It seems the output cvs file doesn't indicate the type of AS event just its coordinates. However, this is crucial for example to efficiently assess AS type frequencies.

3. It would be great if the authors would provide examples of intron retention events as well, which are harder to accurately quantify.

Reviewer 2 Report

This manuscript demonstrated how to  generate the expression level of alternative splicing among two different biological conditions by their program AS-Quant. My major concern is It is not clear what samples were used to generate the data for Figure 2 and 3. Are they the MEF cells used for validation in Figure 4 and 5. If not what is the point of validation. If yes, what are the splicing events of those four genes Ptbp1, Ganab, Camk2g and Tpm3 detected by rMATs and diffSplice.

Other concerns are:

  1. There is no section for methodology. Figure 6 and the description of the calculation on page 9 shall be at the section "Method", not "Discussion".
  2. Titles for Figure 4 and 5 are poor. Shall be more specific to the experiments and its results shown in the figure.
  3. The references are not displaying properly. All are [?]. Authors shall check before and after upload to the system.

Round 2

Reviewer 1 Report

The authors have addressed all concerns raised in a satisfactory manner.

I thus recommend this manuscript for publication in IJMS.

Reviewer 2 Report

The authors addressed my questions.